# Prediction and developing of shear strength of reinforced high strength concrete beams with and without steel fibers using multiple mathematical models

**Ayad Zaki Saber** *

Department of Civil Engineering, Erbil Technical Engineering College, Erbil Polytechnic University, Erbil, Iraq

* ayad.saber@epu.edu.iq

**Data Availability Statement:** All relevant data are within the article and its Supporting Information files.

**Funding:** The authors received no specific funding for this work.

## Abstract

One of the methods to improve the structural design of concrete is by updating the factors given in standard codes, especially when non-conventional materials are used in concrete beams. Accordingly, this study focuses on the colorations between the compressive strength and shear strength of high-strength concrete beams with and without steel fibers. For that purpose, different models are proposed to predict shear strength of high-strength concrete beams, by taking different combinations of the main variables: beam cross-section dimension (width and effective depth), reinforcement index, concrete compressive strength, shear span ratio, and steel fiber properties (volumetric content, fiber aspect ratio, and type of steel fibers). Multi-linear and non-linear regression analyses are used with large database experimental results found in the literature. The predicted results from the proposed equations are composed with different available models from codes, standards, and literatures. The calculated results showed better correlations and were close enough to the experimental data. Based on the data given in the standard codes, the shear strength is proportional to compressive strength ($f'_c$) of the power 0.5. However, this value may not be adequate for modern cement and concrete containing steel fibers. Therefore, the mentioned power value must be reduced 5 times to 0.1.

## 1. Introduction

Shear behavior of structural concrete is complicated, not well-understood, and there is no accepted analytical solution to unify all the factors influencing shear behavior of high-strength reinforced concrete.

Different types of empirical design methods are presented in codes and standards based on the experimental testing of reinforced concrete members [1]. The presented equations do not cover all the factors influencing the shear strength together and there is insufficient testing data for high-strength concrete (HSC) which is considered a relatively new concrete material. A theoretical study is necessary to examine all the existing design equations which are provided in codes, standards, and literature on the shear of HSC beams.

**Competing interests:** The authors have declared that no competing interests exist.

HSC became the most widely used material in the world in recent years, since it reduces the section size, span length and weight of the concrete element.

Classical beam theory provides a simple model for designing beams to resist bending and shear. The prediction of shear is more complicated than bending, and failure may occur in a brittle way. Many different design elements presented to predict shear strength of beams may give non-conservative predictions when applied to HSC beams [2].

Beams mainly fail without advanced warning and developed diagonal cracks that are wider than flexural cracks. To avoid shear failure, an adequate amount of shear reinforcements are used.

Shear strength of reinforced concrete beams depends on several factors including concrete compressive strength $(f_c')$, amount of longitudinal steel, shear span ratio, beam size effect, residual tensile stresses, transmitted directly across cracks and provisions of web reinforcement.

Using of HSC is the tendency of cracks to pass through instead of around the aggregate, this creates smoother crack surfaces and reduces the aggregate interlock, as a result higher dowel forces occur in longitudinal reinforcement bars. These higher dowel forces and highly bond stresses in the HSC beams result higher bond splitting stresses across the shear cracks, which leads to brittle shear failure Using minimum shear reinforcement can control these horizontal splitting cracks and improve shear response [3, 4].

HSC is usually manufactured with a low water to cement ratio (w/c) and has a compressive strength in the range of 50 to 100 MPa. In comparison with normal strength concrete (NSC), the HSC has increased modulus of elasticity, chemical resistance, freeze thaw resistance, lower creep, lower drying shrinkage and lower permeability [5].

Concrete in general has low tensile strength and fails in a brittle manner. The tensile strength of the concrete can be improved by using steel fibers. The cause of low tensile strength is due to the propagation of internal cracks. These internal cracks can be resisted by using steel fiber which are bridging between the two sides of the cracks. As a result, the tensile strength is improved and resist the crack formulation and propagation, this ability avoids the brittle failure mode [6, 7].

## 2. Review of literature

Gomaa & Alnaggar [8] provided an experimental analysis of ultra-high-performance concrete beams failure with varying reinforcement and fiber contents. They showed that the beams failed with shear failure mode for beams without steel fibers and flexural failure mode for beams with steel fiber (2% fiber content). The presence of fibers prevented shear cracks from initiating or developing into critical shear cracks, also ductility is improved.

Al-Nu'man et al. [9] presented an investigation on the strength and deformation characteristics of reinforced NSC and HSC rectangular beams failed in shear and repaired by epoxy injection. They cast and tested ten reinforced concrete beams with dimensions (width b = 100 mm, effective depth d = 180 mm), shear span ratio constant for all beams (a/d = 2.833) and the concrete compressive strength (fc') ranges (21 to 74 MPa), all beams reinforced with same reinforcement ratio (ρ = 0.0131). They concluded that the repaired beams showed a lower stiffness and greater ductility than the original beams. New diagonal shear cracks were developed close to the diagonal shear crack of the original beam before repairing.

Tahenni et al. [10] presented an experimental work on the shear behavior of HSC beams with steel fiber. They cast and tested eight reinforced concrete beams with steel fibers content (0, 0.5, 1 and 2% by volume) and two aspect ratios (65 and 80). The beam dimensions (width b = 100 mm, effective depth d = 136 mm), shear span ratio constant for all beams (a/d = 2.2)

and the concrete compressive strength (fc') is the same for all beams (65 MPa), all beams reinforced with the same reinforcement ratio (ρ = 0.0116). They concluded that the addition of steel fibers does not have an effect on the compressive strength of HSC and does not modify the descending part of the stress-strain curve, but improves the tensile strength of the concrete especially when fibers have a higher aspect ratio. Also, ductility and shear strength were increased. The increase in shear varying between 47% and 88% for steel fiber content of 0.5% to 3%.

Al-Kamal [11] proposed a triangular stress block to predict the nominal flexural strength of HSC beams, the proposed equation and various stress block expressions were examined for estimating the nominal flexural strength of HSC beams using a database of 52 beams available in literature. The theoretical results from this proposed equation showed good and conservative results except some over estimation result for $f'_c = 96\ MPa$, also showed excellent agreement with the results obtained from various codes and proposals by researchers.

Radmila [2] casted and tested 26 normal and high strength concrete reinforced concrete beams with (b = 120 mm) and different values of effective depth d = 180, 240, 300 and 360 mm), shear span ratio (a/d = 1.25, 1.5, 2 and 2.67) and the concrete compressive strength (fc' = 35 MPa) for normal concrete and (fc' = 90 MPa) for high strength concrete, reinforcement ratio ranges (ρ = 0.015 to 0.033). proposed a new equation for prediction of shear strength of HSC beams, based on the experimental researches. The equation is derived on the base that the value of shear strength may be presented as a sum of concrete part of the beam and shear reinforcement contribution. The most important parameters of shear strength are: concrete compressive strength $f'_c$, longitudinal reinforcement (ρ), shear-span ratio $\left(\frac{a}{d}\right)$, and size effect reduction factor. Nonlinear regression analysis was used and tested by using large number (102) of experimental beams results from different literatures. The theoretical results from the proposed equation showed excellent results and good correlation with the experimental test results.

Bao et al. [12] proposed an equation to predict shear strength of HSC beams within a margin of safety and economic design by revising some problems of the ACI-Code shear provision. Three factors: size reduction factor, longitudinal reinforcement ratio (ρ) and arch action factor were introduced by using linear and non-linear regression analysis. The proposed equation was verified with 296 previous experiment testing data found in literatures about HSC and NSC beams. The proposed equation is simpler and more suitable for shear design in practice.

Polak and Dubas [4] presented an investigation on the Canadian code about the influence of concrete compressive strength on the shear strength of HSC beams. A total of 132 existing tests of beams with and without shear reinforcement were analyzed and compared with Canadian code and standards, CAN3-A23.3-M94 and CAN3-A23.3-M84. They concluded that the Canadian code 1994 provisions produce more accurate and less scattered results. Also other parameters such as shear, span ratio $\left(\frac{a}{d}\right)$, longitudinal reinforcement ratio (ρ), and amount of shear reinforcement are effect on the shear strength of the beams in addition to the effect of the compressive strength of the concrete $(f'_c)$.

Reddy et al. [6] presented some improvement on the prediction equation of shear strength of fibrous HSC beams with and without web reinforcement. They cast and tested 16 reinforced concrete beams with dimensions (width b = 100 mm, effective depth d = 130 mm), the concrete compressive strength (fc' = 70 MPa) for all beams, and the same reinforcement ratio is used (ρ = 0.0725). The beams with different shear span ratios $\left(\frac{a}{d} = 1, 2, 3,\ and\ 4\right)$ and various volumetric fiber content (0.4, 0.8 & 1.2%) and aspect ratio (75). The test results indicated an increase in the cracking shear resistance and ultimate shear strength.

Santos et al. [7] presented an experimental study on the beams mode of HSC beams with steel fibers. They concluded that the effectiveness of the fiber reinforcement for the shear resistance is more noticed in shallow beams than in deep beams, also they proposed some models and recommended (RILEM TC 162 TDF) for the prediction of the fiber reinforcement contribution in the shear resistance of concrete beams.

Radmila [13] presented a study on the relationship between the compressive strength of concrete and the tensile strength and shear strength of HSC beams. The experimental researches of HSC properties show that most relations are valid for NSC and may not be applied on HSC. Experimental results were done on HSC (C90/C150) and NSC (C35/C45). The results indicated that the concrete compressive strength does not contribute significantly to the increase of inclined cracking strength of reinforced concrete beams, and special attention is necessary for estimating the concrete contribution to shear strength of HSC beams.

Radmila [14] presented an experimental study on the effect of arch action on the shear strength of HSC beams, the beam was made of NSC (C40 & C50) and HSC (C90 & C105) and shear-span ratio $\left(\frac{d}{a}\right)$ varies from 1.25 to 2.69, also the effect of longitudinal reinforcement ($\rho$) is considered. The experimental results indicated that the arch action effect appear in beams with $\left(\frac{a}{d} < 2.5\right)$. By decreasing the shear-span ratio $\left(\frac{a}{d}\right)$, the effect of arch action is increased and it increases more in the case of HSC than in the case of NSC.

Yuan et al. [15]; presented an experimental study on the shear capacity contribution of steel fiber on the shear strength of HSC beams with and without stirrups. Five large-scale HSC beams with a steel fiber content of 0.75% by volume were tested. The results indicated that all beams failed by a shear-activated failure and using of 0.75% steel fiber, increase the shear strength by 13.2% compared with that of minimum shear reinforcement for the HSC beam. The prediction models containing fiber factors were more closely in agreement with test results with a minimum of 10.9% difference.

Saghair et al. [16] presented an experimental study on shear behavior of HSC beams by testing 19 beams with dimensions (width b = 120 mm and effective depth d = 275 mm), shear span ratio (a/d = 1, 2, 3 and 4) are used, concrete compressive strength (fc' = 50 and 80 MPa) for high strength concrete and (30 MPa) for normal concrete beams and the reinforcement ratio ($\rho$ = 0.00515) for all beams. and compared with NSC, the concrete compressive strength ranged from 30 to 80 MPa, they included the effect of shear-span ratio $\left(\frac{a}{d}\right)$, amount of shear reinforcement (stirrup spacing S) and the angle of inclination of the stirrups. They proposed equations of predicting cracking, ultimate shear strength, and the ratio between them for both HSC and NSC.

Mphonde & Frantz [17] presented an experimental study by testing 12 beams with a constant shear-span ratio $\frac{a}{d} = 3.6$ and concrete compression strength vary from 21 to 84 MPa. The beamwidth b = 150 mm, effective depth d = 298.5 mm and longitudinal reinforcement ratio (0.0329). They concluded that the ACI-code shear design method to be very conservative, also they presented a new equation to more accurately predict the ultimate shear strength of HSC concrete beams.

Fujita et al. [18]; presented an experimental investigation on the shear strength of high-strength reinforced concrete beams with concrete compressive strength ranging 36–100 MPa and the shear-span ratio of 2–5, different beam dimensions (b, d) are used (150, 250), (150, 500) and (350, 1000) mm and longitudinal reinforcement ratio ranges (0.0153 to 0.0136). They concluded that the size effect is more prominent in HSC than in NSC beams. New empirical equations are proposed for calculating the shear strength of high-strength reinforced concrete beams without shear reinforcement, and the concrete compressive strength ranging of 80–125 MPa.

Kolhapure [19] presented an experimental investigation on twelve HSC beams of dimensions (b = 125 mm and d = 100 mm) and concrete compressive strength (65 MPa), including the effect of shear-span ratio (a/d = 3 and 5), longitudinal reinforcement ratio ranges (0.008 to 0.032) and minimum web reinforcement. The experimental results are compared with different codas equations, Indian code of practice, ACI-Code 318 & BS8110, the British code model is proposed for predicting shear strength.

Sarsam and Al-Musawi [20] tested (14) beams with stirrups, the experimental results are used with additional 107 data from the literature to determine shear strength of the HSC & NSC beams using different available design equations from codes and standards. They showed that ACI-code equation gives overestimated results for both HSC and NSC over a wide range of the concrete compressive strength ($f_c'$), longitudinal reinforcement ratio ($\rho$), stirrup minimal strength ($\rho_v \cdot f_{yv}$), and shear-span ratio. Also New Zealand code & British code methods are less conservative than ACI-code method.

Motamed et al. [21] presented shear design equations for beams with horizontal web bars which is provide a more accurate prediction of shear strength of high-strength concrete beams. Also, the experimental results showed that the shear strength of HSC beams are highly dependent on dowel action resulting from web bars.

Roller and Russel [22]; presented an experimental investigation on the shear strength of HSC beams with web reinforcement. The concrete compressive strength of the beams was 69, 117 & 124 MPa. They concluded that for non-prestressed HSC members subject to shear and flexure only the minimum quantity of shear reinforcement specified in ACI-code needs to increase as the concrete compressive strength increases.

Ashour et al. [23] tested 18 rectangular high-strength fiber reinforced concrete beams with dimensions (b = 125 mm and d = 215 mm), the beams are subjected to combined flexure and shear. The main variables were the steel fiber content (0.5 and 1%) by volume and aspect ratio (75), the longitudinal steel reinforcement ratio ranges (0.0284 to 0.0458) and the shear-span ratio (1, 2, 5 and 6), the concrete compressive strength ranges (92 to 101 MPa). They proposed two empirical equations to predict the shear strength of high-strength fiber reinforced concrete beams without shear reinforcement, the results gave good predictions for the shear strength of the tested beams, also the addition of steel fiber increased the beam stiffness and ductility.

Valle and Buyukozturk [24] investigated the shear strength and ductility of fiber-reinforced HSC under direct shear. Two types of fibers (steel fiber and polypropylene fiber) with or without stirrups were used. They concluded that the addition of fibers increased the shear strength by about 60% with steel fiber and 17% with polypropylene, also shear deformation and ductility are improved and the relative toughness was greater about 5 times than the plain concrete specimens. They proposed an equation to predict the shear strength which correlates well with the experimental results, also shear stresses and shear strains are predicted with good accuracy.

Existing major shear design formulae established primarily for conventional concrete beams were assessed for recycled aggregate concrete (RAC) beams. Results showed that when applied to the shear test database compiled for RAC beams, those formulae provided only inaccurate estimations with surprisingly large scatter. To cope with this bias, machine learning (ML) techniques deemed as potential alternative predictors were resorted to. First, a Grey Relational Analysis (GRA) was carried out to rank the importance of the parameters that would affect the shear capacity of RAC beams. Then, two contemporary ML approaches, namely, the artificial neural network (ANN) and the random forest (RF). It was found that both models produced even better predictions than the evaluated formulae. With this superiority, a parametric study was undertaken to observe the trends of how the parameters played roles in influencing the shear resistance of RAC beams. The findings indicated that, though

less influential than the structural parameters such as shear span ratio, the effect of the replacement ratio of recycled aggregate (RA) was still significant, for safe application of RAC, using partial factors calibrated to consider the uncertainty is feasible when designing the shear strength of RAC beams [25].

The use of recycled concrete aggregates as an alternative aggregate material in concrete, where natural aggregates are replaced with recycled concrete aggregates, is a promising technology for conserving natural resources and reducing the environmental impact of concrete. Xie et al. [26] presents a study on mechanical and durability properties of concretes manufactured with recycled aggregates of different sizes and contents. A total of 14 batches of RACs were manufactured. The compressive strength, elastic modulus, flexural strength, splitting tensile strength, workability, drying shrinkage, and water absorption of each batch are experimentally tested. Test following parameters are considered (recycled aggregate replacement ratio, size of coarse aggregates, and mixing method used in the preparation of concrete). The results indicate that the compressive strength is an important factor on mechanical and durability-related properties of RACs in addition to the properties of different RAC mixes of the same compressive strength such as (the size and content of the coarse aggregates).

Based on the results given in the literature [3, 12, 27–57], it can be said that most of the models were underestimated the average ratio of ($R_{avg}$) that have been found based on the experimental to predicted shear ($R_{avg}>1.0$). For example, the models are given by the ACI-Code [27, 29], Kim and Wight [30], CEP-FIP 1990 [34] and 1993 [35], CSA94 [36], EHE 99 [39], EC2002 model [40], AS3600 model [41], IS456:2000 model [43], Arsalan [45], JSCE 1996 [47], and Bazant & Kim model [48], where the ($R_{avg}$) ranged between (1.142 and 2). While the results of other models are overestimated, the average ratio ($R_{avg}<1.0$). For example, the models are proposed by Hammad et al. model [3], Zsulty model [37], NZS3101 model [42], Niwa et al. model [49], Bazant & Kim model [50], the ($R_{avg}$) value ranged between (0.637 and 0.922). Nevertheless, the following models showed acceptable results of shear strength such as ACI-Code model [28], Russo et al. model [31], Bae et al. model [12], CEP-FIP model [33] and Gastebled & May model [38], where the ($R_{avg}$) value near the unity but they are still not close to one.

The shear behavior of HSC is complicated and not well-understood, that is why more research is necessary to study the behavior of HSC with and without steel fiber and to find the best equation to predict the shear strength ($V_c$). As mentioned above, most of the standards, codes equations and recommendations have been overestimated/underestimated shear strength. In addition, the shear strength is proportional to ($f'_c$) of the power (0.5) which is overestimated, that is why the power of ($f'_c$) is investigated in this study and new equations with $R_{avg}$ equivalent to 1 has been proposed.

The new participation of this study is including the fiber factor (F) which represents the fiber properties (fiber content $Q_f$%, fiber aspect ratio $\frac{L}{d}$, and fiber type) in the equation of shear strength in addition to other variables considered in this study.

## 3. Research significance

Most of the codes equations and recommendations give overestimated shear strength results and because of the complicated behavior of shear, this study is presented to suggest different models including different variables that affect the shear strength of HSC beams with and without steel fiber, such as beam cross-section dimensions, beamwidth ($b_w$), and effective width ($d$), shear span ratio $\left(\frac{a}{d}\right)$, concrete compressive strength ($f'_c$), flexural reinforcement ratio ($\rho$), and steel fiber properties (fiber content $Q_f$%, fiber aspect ratio $\frac{L}{d}$, and fiber type).

## 4. Methodology

In order to get the objective of this study, specifically, prediction of shear strength of HSC beams with and without steel fibers, 184 experimental tests (**S1 Table**) were collected in previous literature for HSC beams, and 61 experimental tests (**S2 Table**) were collected for beams with steel fiber. All these datasets were then used to predict the mentioned model.

Additionally, different types of models are proposed including all possible combinations of the variables mentioned in Section 3, then the best model is recommended to predict shear strength of HSC beams with and without steel fiber, also the results are compared with different equations found in available codes and recommendations. In order to ensure the authenticity and reliability of these data, four strict criteria were selected, i.e. (i) the main focus was made on the data given by international standards and codes (practical guidance), then scientific papers; (ii) the selected study must be published in the reliable journal, especially Scopas and Clarivate; (iii) The selected data must be related to the high strength concrete; (iv) All selected studies must have enough data to find the coefficient of correlation (r), standard deviation ($\sigma$), variation, $R_{average}$, $R_{maximum}$ and $R_{minimum}$. In addition, the experimental results of previous studies showed that the shear strength of concrete beams ($v_c$) depends on the concrete compressive strength ($f_c'$), longitudinal reinforcement index ($\rho$), and shear span ratio $\left(\frac{a}{d}\right)$. The available models of predicting ($v_c$) from the codes and references also depend on these parameters, that is why these parameters are selected from the literatures and previous studies to propose new models for predicting shear strength of high strength concrete beams.

## 5. Theoretical analysis

### 5.1 Non-linear regression model

Different types of models are proposed including all possible effecting variables, such as beam cross-section dimensions (width $b_w$ and effective width $d$), concrete compressive strength ($f_c'$), flexural reinforcement ratio ($\rho_l$), shear span ratio $\left(\frac{a}{d}\right)$, and steel fiber properties (fiber content $Q_f$, fiber aspect ratio $\frac{L}{d}$, and fiber type represented by bond factor).

The main non-linear regression equation in the following form:

$$V_c = \alpha_0 \cdot b_w^{\alpha_1} \cdot d^{\alpha_2} \cdot f_c^{\alpha_3} \cdot \left(\frac{a}{d}\right)^{\alpha_4} \cdot (\rho_l)^{\alpha_5} \cdot F^{\alpha_6} \qquad 1$$

The coefficient ($\alpha_0$ to $\alpha_6$) are constant and determined using nonlinear regression analysis method and principle of least square, by using (184) experimental tests found in literature and shown in **S1 Table** for HSC beams without steel fiber and (61) tests for HSC beams with steel fiber and shown in **S2 Table**.

As a result, the following models are proposed:

$$V_c = \left[4.2 \times f_c'^{0.133} \cdot \rho_l^{0.274} \cdot \frac{d^{0.618}}{a}\right] b_w d \qquad 2$$

$$V_c = \left[0.464 \times f_c'^{0.11} \cdot \left(\rho_l \cdot \frac{d}{a}\right)^{0.41}\right] b_w d \qquad 3$$

$$V_c = [0.88 f_c'^{0.103}] b_w d \qquad 4$$

Figs 1–3 show the relationship between the predicted results of the Eqs 2, 3 and 4 respectively versus the experimental data. As shown the results are close enough and in good

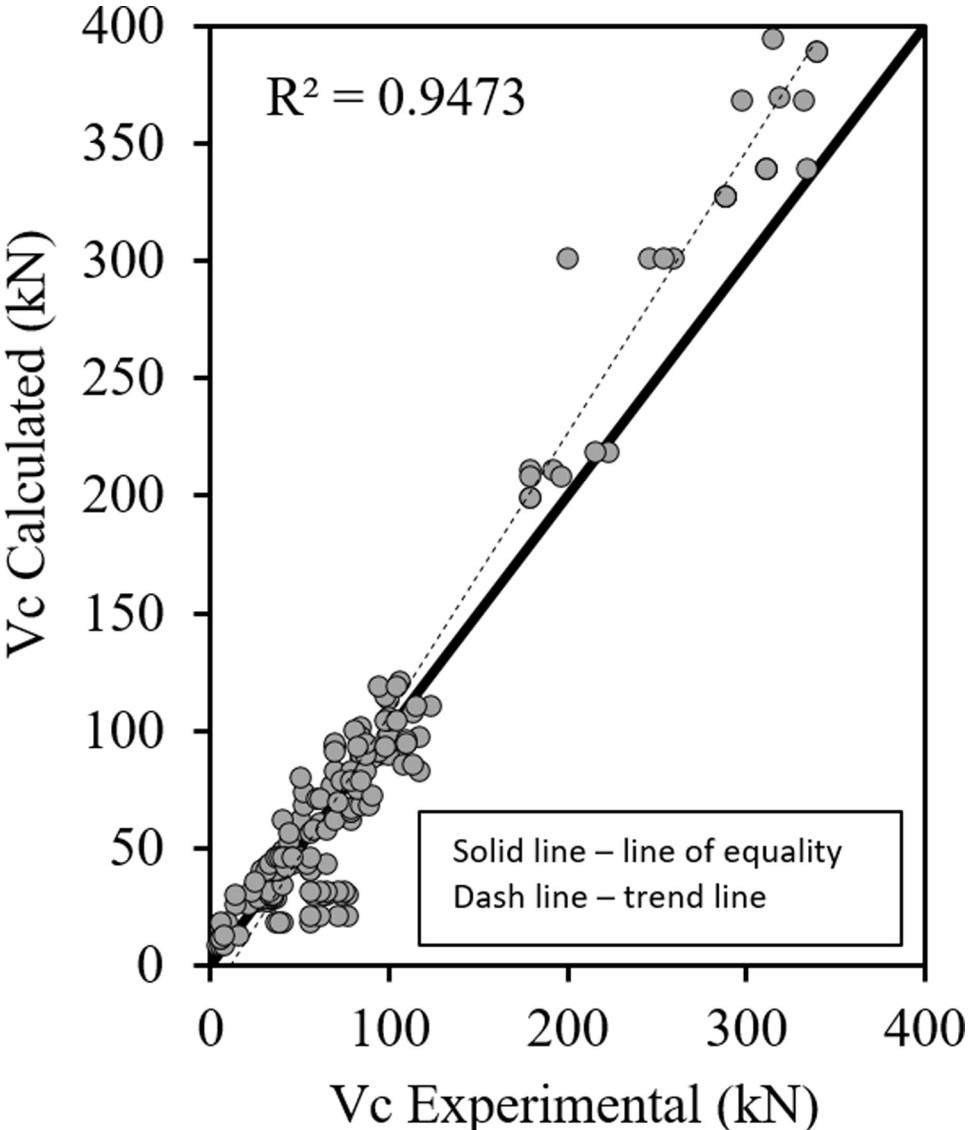

**Fig 1. Shear strength of HSC beams (Eq 2).**

correlation, and all data points are near the middle line and the statistical results are shown in Table 1.

## 5.2 ACI-code models

Other models are proposed using the shear strength by ACI-code equation $\left( V_c = \frac{\sqrt{f_c'}}{6} b_w d \right)$ as a base value, then correction or modified factor is proposed for HSC as the following:

$$V_c = \phi V_{c\ ACI} \qquad\qquad 5$$

The shear modification factor is determined as a function of the same variables mentioned before, using the same nonlinear regression analysis and same experimental data, the following

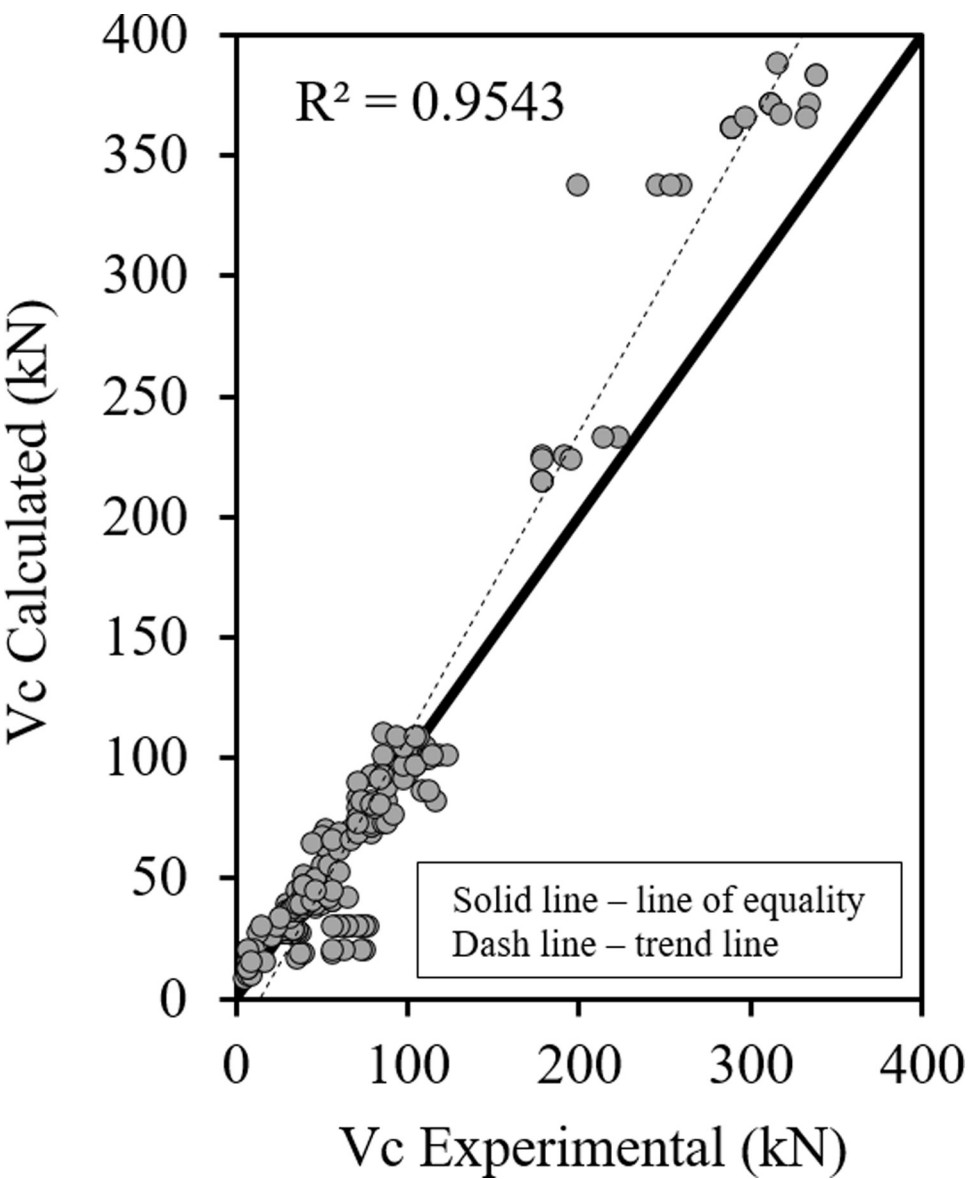

**Fig 2. Shear strength of HSC beams (Eq 3).**

equations are proposed:

$$V_c = \left[ 5.526 \rho_l^{0.252} \cdot \frac{d^{0.664}}{a} \right] \frac{\sqrt{f_c'}}{6} b_w d \qquad \qquad 6$$

$$V_c = \left[ 8.32 (\rho_l \cdot \frac{d}{a})^{0.415} \right] \frac{\sqrt{f_c'}}{6} b_w d \qquad \qquad 7$$

The quantity between the brackets represents the shear modification factor ($\phi$) which is a function of different variables.

Figs 4 and 5 show the relationship between the predicted results of the Eqs 6 and 7 respectively versus the experimental data, as shown the results are close enough and in good correlation, also all data points are near the middle line and the statistical results are shown in Table 2.

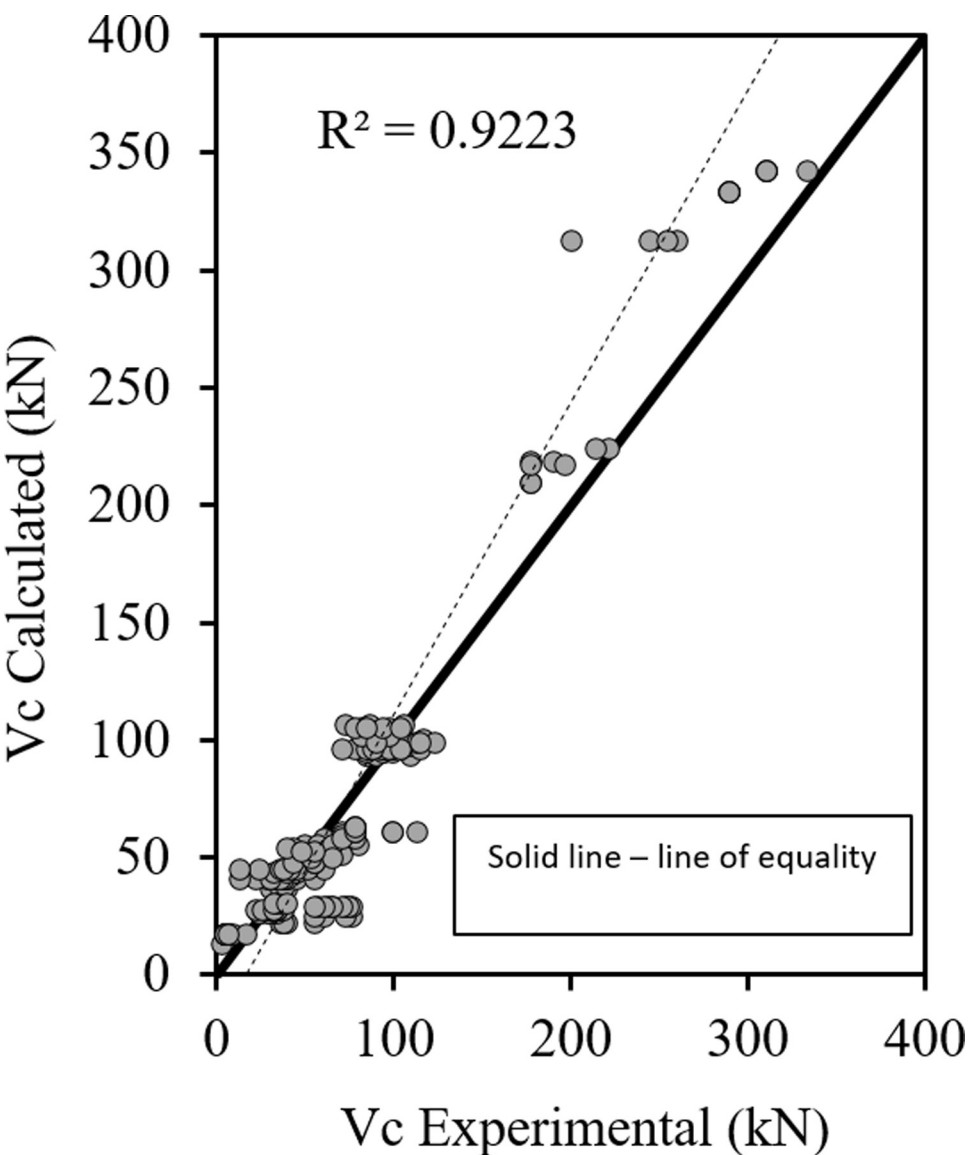

**Fig 3. Shear strength of HSC beams (Eq 4).**

## 5.3 Multilinear regression model

In the following section, multilinear regression analysis method is used to propose theoretical equations to predict shear strength of HSC beams, the proposed model taking the following

**Table 1. Statistical results used in Eqs 2–3.**

| Equation | Coeff. of correlation (r) | Variance (var.) | Standard deviation ($\sigma$) | Average $R = \frac{V_{exp}}{V_{calc}}$ | Max R | Min R |
|---|---|---|---|---|---|---|
| 2 | 0.973 | 0.25 | 0.5 | 1.08 | 3.59 | 0.332 |
| 3 | 0.977 | 0.271 | 0.521 | 1.086 | 3.73 | 0.302 |
| 4 | 0.96 | 0.25 | 0.5 | 1.088 | 3.11 | 0.304 |

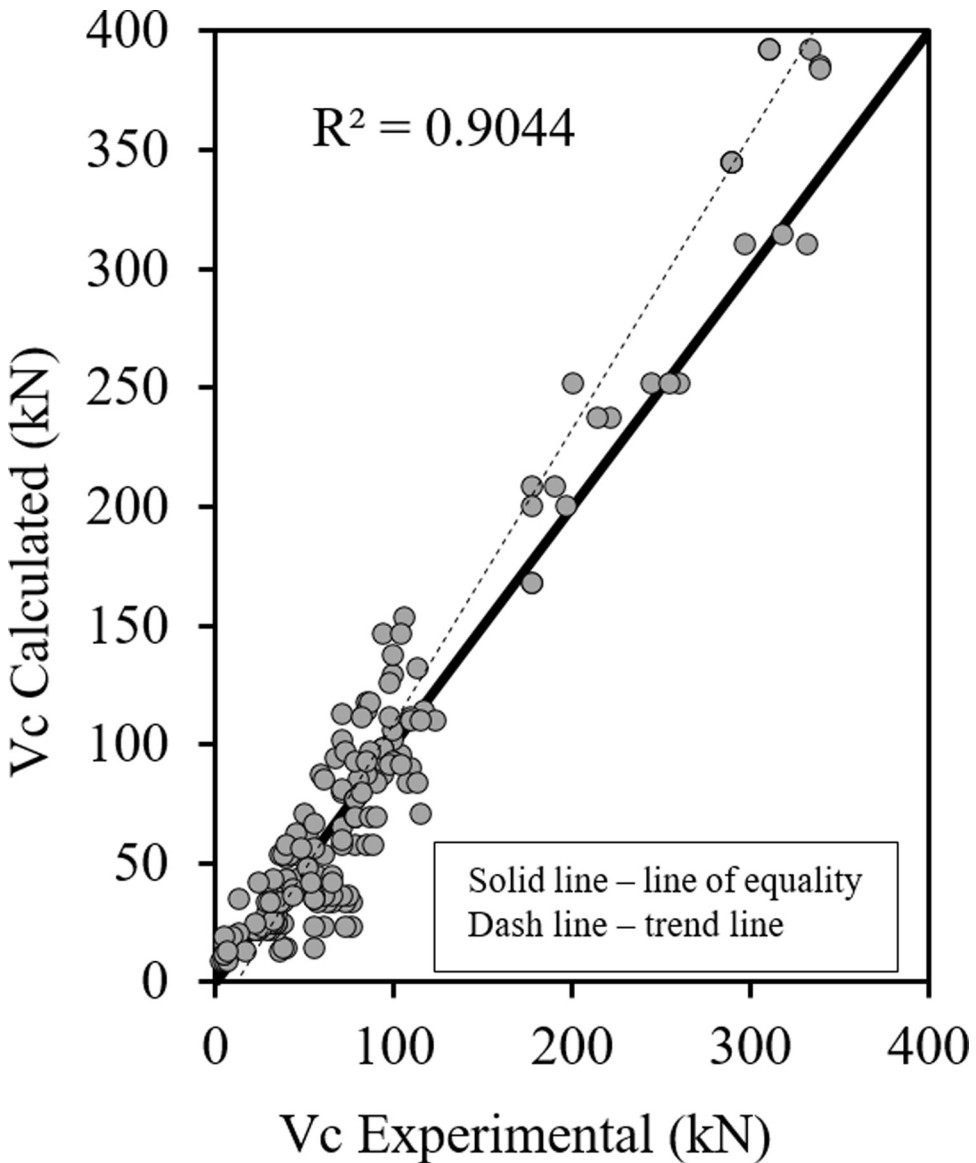

**Fig 4. Shear strength of HSC beams (Eq 6).**

form (Eqs 8–9).

$$V_c = \left[ 0.145\sqrt{f_c'} + 46.776\rho_l \cdot \frac{d}{a} \right] b_w d \qquad\qquad 8$$

$$V_c = \left[ 34.263\rho_l + 1.39\frac{d}{a} \right] \frac{\sqrt{f_c'}}{6} b_w d \qquad\qquad 9$$

Figs 6 and 7 show the relationship between the predicted results of the Eqs 8 and 9 respectively versus the experimental data, as shown the results are close enough and in good correlation, also all data points are near the middle line and the statistical results are shown in Table 3.

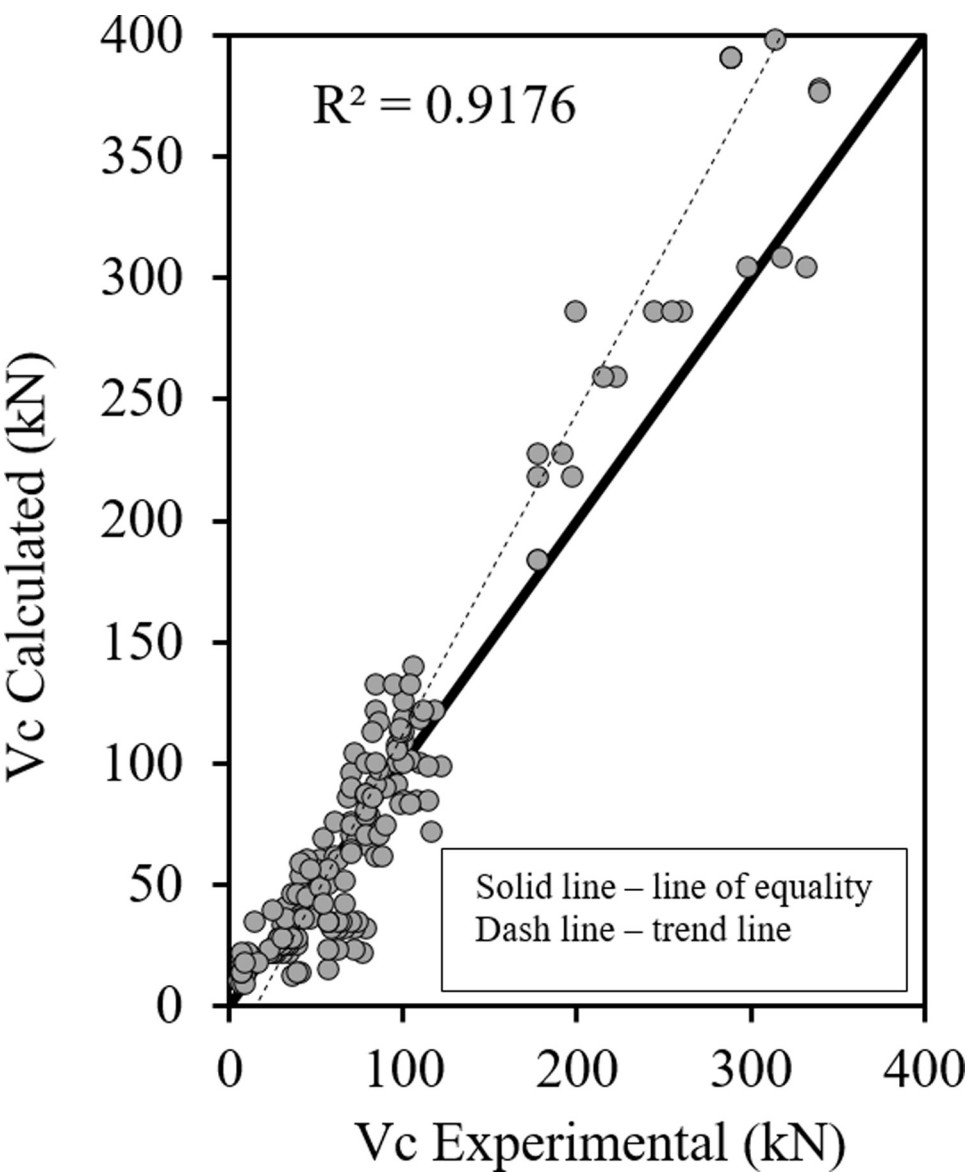

**Fig 5. Shear strength of HSC beams (Eq 7).**

## 6. Optimization and developing models

In this section, the output of the above three models was compared with the models given by the literature review. All information regarding this matter has been summarized in Table 4 in order to help readers and easily get show the information without using long paragraphs. Thus, the predicted results from these proposed models are compared with the (26) available models, codes, and recommendations, the statistical results are shown in Table 4.

**Table 2. Statistical results used in equations 6–7.**

| Equation | r | Var. | $\sigma$ | $R_{avg}$ | $R_{max}$ | $R_{min}$ |
|---|---|---|---|---|---|---|
| 6 | 0.951 | 0.273 | 0.523 | 1.09 | 3.729 | 0.319 |
| 7 | 0.958 | 0.3 | 0.547 | 1.095 | 3.874 | 0.282 |

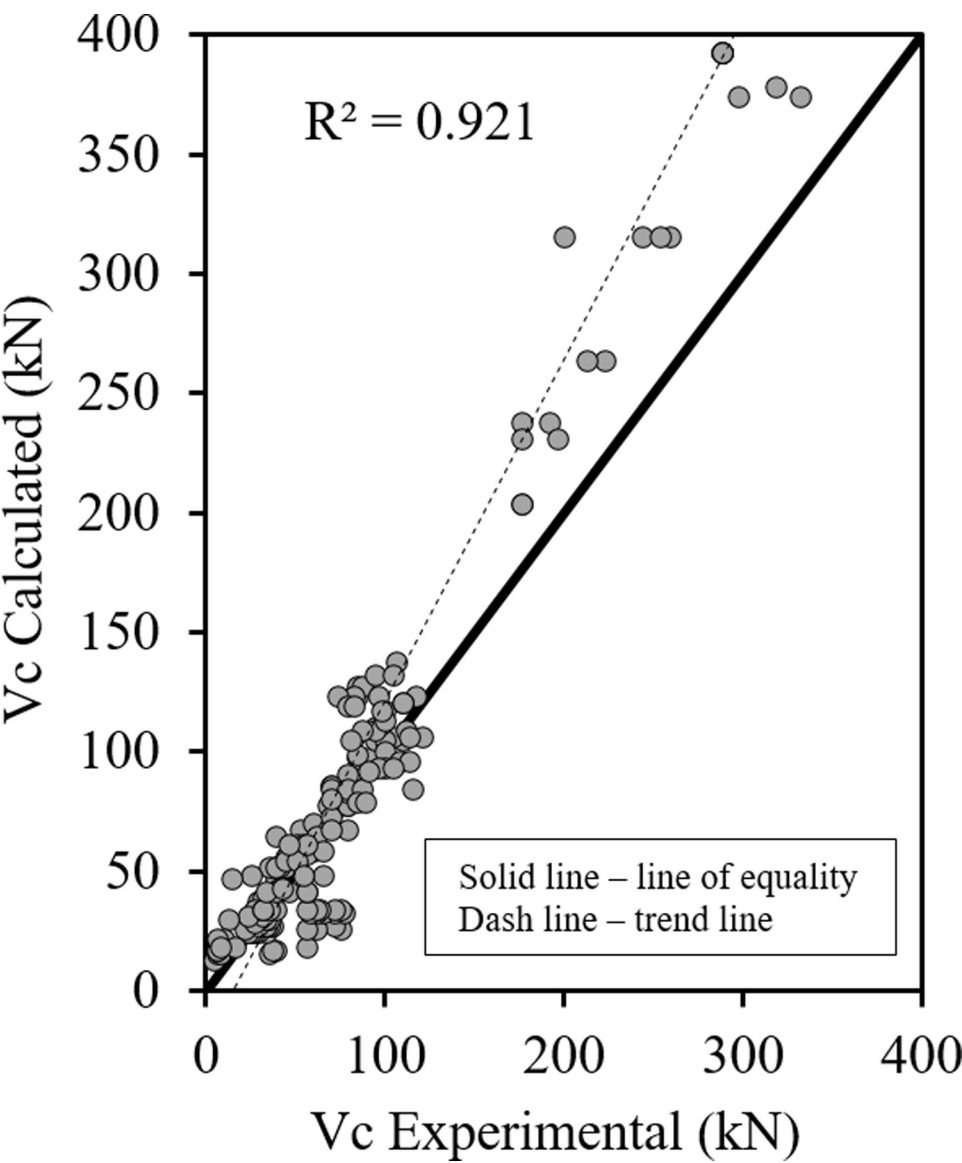

**Fig 6. Shear strength of HSC beams (Eq 8).**

In the second part of this study, the shear strength of HSC with steel fiber is predicted by using the same previous method which is used for a beam without steel fiber, new factor is included which presents the properties of the steel fiber.

$$F = Q_f \cdot \frac{L}{d} \cdot d_f \qquad\qquad 10$$

Where $Q_f$ = steel fiber content (% volumetric percent), $\frac{L}{d}$ = aspect ratio of steel fiber, $L$ = Length (mm), $d$ = diameter (mm), $d_f$ = bond factor (depend on the type of steel fiber), value ranges between 0.9 and 1.2 [58, 59].

The following equations are proposed:

$$V_c = \left[ 0.89(1+F)f_c'^{0.91} \left( \frac{V_u \cdot d}{M_u} \right) (\rho_l)^{0.67} \right] b_w d \qquad\qquad 11$$

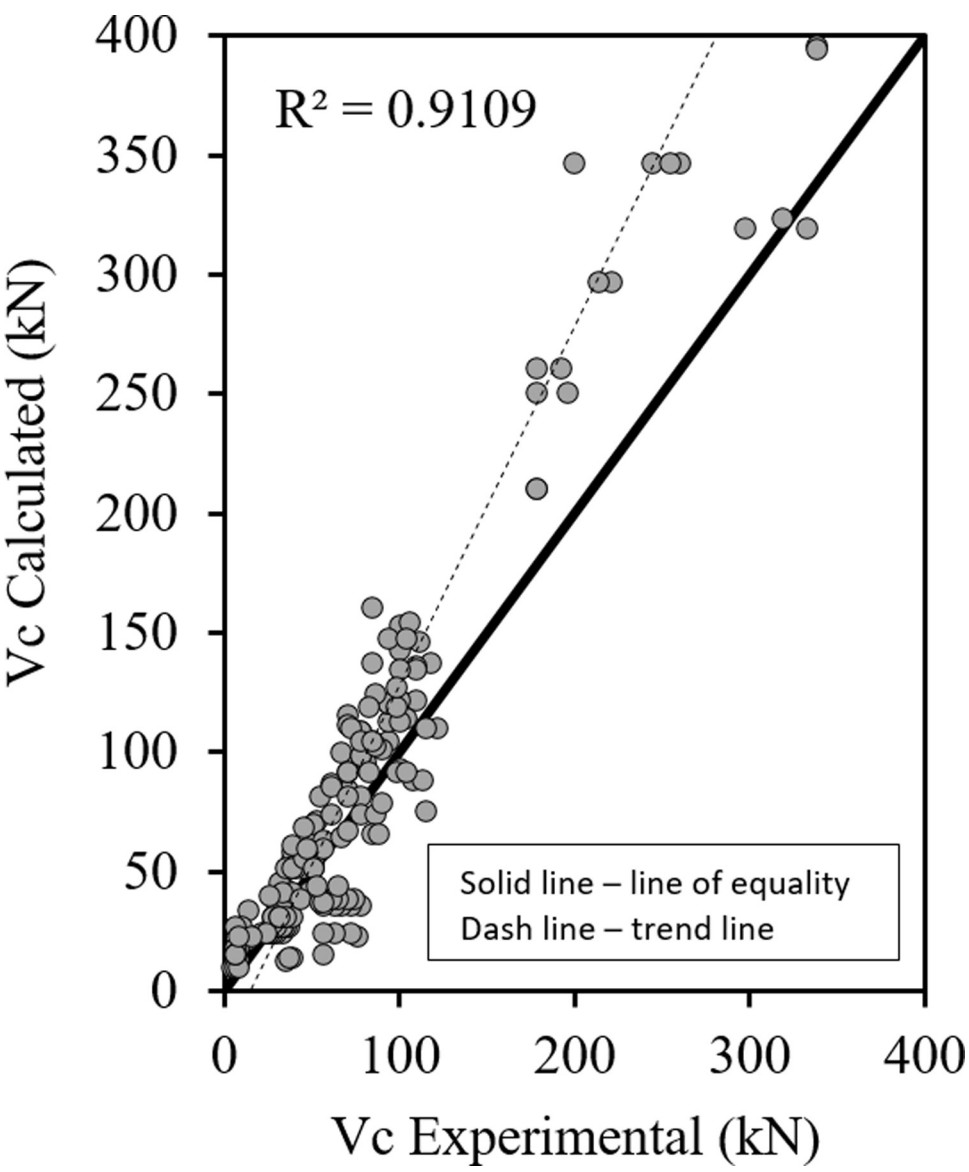

**Fig 7. Shear strength of HSC beams (Eq 9).**

$$V_c = \left[1.46(1+F)f_c'^{0.814}\left(\rho_l \cdot \frac{V_u \cdot d}{M_u}\right)^{0.756}\right]b_w d \qquad 12$$

**Table 3. Statistical results used in Eqs 8–9.**

| Equation | r | Var. | $\sigma$ | $R_{avg}$ | $R_{max}$ | $R_{min}$ |
|---|---|---|---|---|---|---|
| 8 | 0.96 | 0.224 | 0.473 | 1.0 | 3.11 | 0.287 |
| 9 | 0.954 | 0.279 | 0.528 | 0.984 | 3.692 | 0.229 |

**Table 4. The range of the experimental data and range of application of the mathematical proposed equations.**

| No. | Available equation (model) * | Reference | r | St. dv. $\sigma$ | Var. | $R_{avg}\,\frac{V_{c\,exp}}{V_{c\,cal}}$ | $R_{max}$ | $R_{min}$ |
|---|---|---|---|---|---|---|---|---|
| 1 | ACI-Code equation: $V_c = \frac{1}{6}\sqrt{f_c'}\,b_w d$ | [27] | 0.94 | 0.544 | 0.296 | 1.179 | 3.466 | 0.282 |
| 2 | ACI-Code equation: $V_c = \left[0.16\sqrt{f_c'} + 7\rho_l \cdot \frac{V_u d}{M_u}\right]b_w d$ | [28] | 0.949 | 0.506 | 0.256 | 1.096 | 3.31 | 0.285 |
| 3 | ACI-Code equation: $V_c = \left[\sqrt{f_c'} + 120\rho_l \frac{V_u d}{M_u}\right]b_w d$ | [29] | 0.95 | 0.56 | 0.313 | 1.21 | 3.67 | 0.312 |
| 4 | Kim & Wight: $V_c = 0.2\left(1-\sqrt{\rho}\right)\left(\frac{a}{d}\right)^r\left[\sqrt{f_c'} + 1020\rho\left(\frac{d}{a}\right)^{0.6}\right]b_w d$ | [30] | 0.97 | 0.817 | 0.67 | 1.652 | 5.25 | 0.411 |
| 5 | Russo et al.: $V_c = \xi\left[0.97\rho_l^{0.46}\sqrt{f_c'} + 0.2\rho^{0.91}f_c'0.38f_{yl}^{0.96}\left(\frac{a}{d}\right)^{-\frac{5}{2}}\right]b_w d$ | [31] | 0.939 | 0.472 | 0.222 | 1.049 | 3.09 | 0.328 |
| 6 | Bazant & Kim: $V_c = \xi\left[0.83\rho_l^{\frac{1}{3}}\sqrt{f_c'} + 206.9\rho^{\frac{5}{6}}\left(\frac{a}{d}\right)^{-\frac{5}{2}}\right]b_w d$ | [32] | 0.925 | 0.372 | 0.138 | 0.765 | 2.474 | 0.123 |
| 7 | Bae et al.: $V_c = \left[4.65\left(5.5-1.5\frac{a}{d}\right)\frac{\lambda\sqrt{f_c'}\rho^{0.25}}{(1+d)^{0.4}}\right]b_w d$ | [12] | 0.872 | 0.573 | 0.328 | 0.949 | 3.924 | 0.315 |
| 8 | CEP-FIP: $V_c = [0.15\left(1+\sqrt{\frac{200}{d}}\right)\left(\frac{3d}{a}\right)^{\frac{1}{3}}(100\rho_l f_c')]^{\frac{1}{3}}b_w d$ | [33] | 0.97 | 0.435 | 0.189 | 1.064 | 3.026 | 0.224 |
| 9 | CEP-FIP 1990: $V_c = [0.12\left(1+\sqrt{\frac{200}{d}}\right)\left(\frac{3d}{a}\right)^{\frac{1}{3}}(100\rho_l f_c')]^{\frac{1}{3}}b_w d$ | [34] | 0.97 | 0.544 | 0.295 | 1.33 | 3.783 | 0.28 |
| 10 | CEP-FIP 1993: $V_c = [0.12\left(1+\sqrt{\frac{200}{d}}\right)(f_c')^{\frac{1}{3}}(100\rho_l)]^{\frac{1}{3}}b_w d$ | [35] | 0.968 | 0.585 | 0.343 | 1.404 | 3.855 | 0.28 |
| 11 | CSA 94: $V_c = \lambda\sqrt{f_c'}\,b_w d$ <br> $\lambda = 0.2$ for $d = 300mm$ & $\lambda = \left(\frac{220}{1000+d}\right)$ for $d > 300mm$ | [36] | 0.947 | 0.445 | 0.198 | 1.119 | 2.89 | 0.235 |
| 12 | Zsutty: $V_c = \alpha\left[2.1746\left(f_c' \cdot \rho \cdot \frac{d}{a}\right)\right]^{\frac{1}{3}}b_w d$ <br> $\alpha = 1.0$ for $\frac{a}{d} \geq 2.5$ & $\alpha = 2.5\frac{d}{a}$ for $\frac{d}{a} \leq 2.5$ | [37] | 0.955 | 0.436 | 0.19 | 0.831 | 2.87 | 0.25 |
| 13 | Hammad et al.: $V_c = \left[1.05\left(f_c' \cdot \rho_l \cdot \frac{d}{a}\right)\right]^{0.38}b_w d$ | [3] | 0.95 | 0.312 | 0.097 | 0.637 | 1.785 | 0.154 |
| 14 | Gastebled & May: $V_c = \left[0.15\left(\frac{37.41}{\sqrt{d}}\right)\left(\frac{3d}{a}\right)^{\frac{1}{3}}(100\rho_l)^{\frac{1}{6}}(1-\sqrt{\rho})^{\frac{2}{3}}f_c'0.35\right]b_w d$ | [38] | 0.966 | 0.376 | 0.141 | 1.048 | 2.357 | 0.186 |
| 15 | EHE 99 Spanish: $V_c = [0.12\xi(100 \cdot \rho_l \cdot f_c')]^{\frac{1}{3}}b_w d$ | [39] | 0.953 | 1.213 | 1.471 | 2 | 7.92 | 0.56 |
| 16 | EC 2002: $V_c = \left[0.18\left(1+\sqrt{\frac{2}{d}}\right)(100\rho_l \cdot f_c')\right]^{\frac{1}{3}}b_w d$ | [40] | 0.967 | 0.585 | 0.342 | 1.4 | 3.855 | 0.28 |
| 17 | AS3600: $V_c = [\beta_1\beta_2\beta_3(\rho_l \cdot f_c')]b_w d$ <br> $\beta_1 = 1.1\left(1.6-\frac{d}{1000}\right); \beta_2 = 1.0; \beta_3 = 2\frac{d}{a}$ | [41] | 0.945 | 0.589 | 0.344 | 1.35 | 4 | 0.342 |
| 18 | NZS 3101: $V_c = (0.07 + 10\rho_l)\sqrt{f_c'}\,b_w d$ | [42] | 0.932 | 0.452 | 0.208 | 0.778 | 3.062 | 0.147 |
| 19 | IS456:2000: $V_c = \left[\left(0.85\sqrt{0.8f_c'}\right)\frac{(\sqrt{1+5\beta}-1)}{6\beta}\right]b_w d,\ \beta = \frac{0.8f_c'}{689\rho_l}$ | [43] | 0.976 | 0.792 | 0.628 | 1.635 | 5.326 | 0.43 |
| 20 | BS8101: $V_c = \left[\frac{0.79}{\gamma_m}(100\rho_l)^{\frac{1}{3}}\left(\frac{f_{cu}}{25}\right)^{\frac{1}{3}}\left(\frac{400}{d}\right)^{\frac{1}{4}}\right]b_w d$ | [44] | 0.953 | 0.347 | 0.12 | 0.922 | 1.837 | 0.087 |
| 21 | Arsalan 2008: $V_c = [0.12\sqrt{f_c'} + 0.02f_c'0.65]b_w d$ | [45] | 0.936 | 0.585 | 0.342 | 1.258 | 3.772 | 0.296 |
| 22 | Russo et al. 2013: $V_c = \lambda\left[\rho^{0.4} \cdot f_c'0.39 + 0.5\rho^{0.83}f_y^{0.89}\left(\frac{a}{d}\right)^{-1.2-0.45\frac{a}{d}}\right]b_w d$ | [46] | 0.96 | 0.403 | 0.162 | 0.967 | 2.812 | 0.272 |
| 23 | JSCE 1996: $V_c = \left[0.2f_c'^{\frac{1}{3}}(100\rho_l)^{\frac{1}{3}}\left(\frac{1000}{d}\right)^{\frac{1}{4}}\right]b_w d$ | [47] | 0.967 | 0.468 | 0.219 | 1.142 | 3.095 | 0.228 |
| 24 | Bazant & Kim: $V_c = \lambda\left[8\rho^{\frac{1}{3}}\left(\sqrt{f_c'} + 300\sqrt{\frac{\rho_l}{\left(\frac{a}{d}\right)}}\right)\right]b_w d$ | [48] | 0.956 | 0.59 | 0.348 | 1.341 | 4.127 | 0.3 |
| 25 | Niwa et al.: $V_c = \left[0.2f_c'^{\frac{1}{3}}(100\rho_l)^{\frac{1}{3}}\left(\frac{1000}{d}\right)^{\frac{1}{2}}\right]\lambda b_w d$ | [49] | 0.97 | 0.357 | 0.127 | 0.872 | 2.487 | 0.189 |
| 26 | Bazant & Kim: $V_c = \left[0.52 + \rho_l^{\frac{1}{3}}\left(\sqrt{f_c'} + 249\sqrt{\frac{\rho_l}{\left(\frac{a}{d}\right)}}\right)\frac{\left(1+\sqrt{\frac{5.08}{d_a}}\right)}{\sqrt{\left(4+\frac{d}{25d_a}\right)}}\right]b_w d$ | [50] | 0.925 | 0.38 | 0.145 | 0.782 | 2.53 | 0.189 |
| **Average** | | | **0.95** | **0.53** | **0.31** | **1.15** | **3.48** | **0.27** |

* Concrete compression strength $(f_c') = 20 - 90$ Mpa; Flexural longitudinal reinforcement $(\rho\%) = 1.3 - 3.3\%$; Shear span ratio $\left(\frac{a}{d}\right) = 2 - 5$.

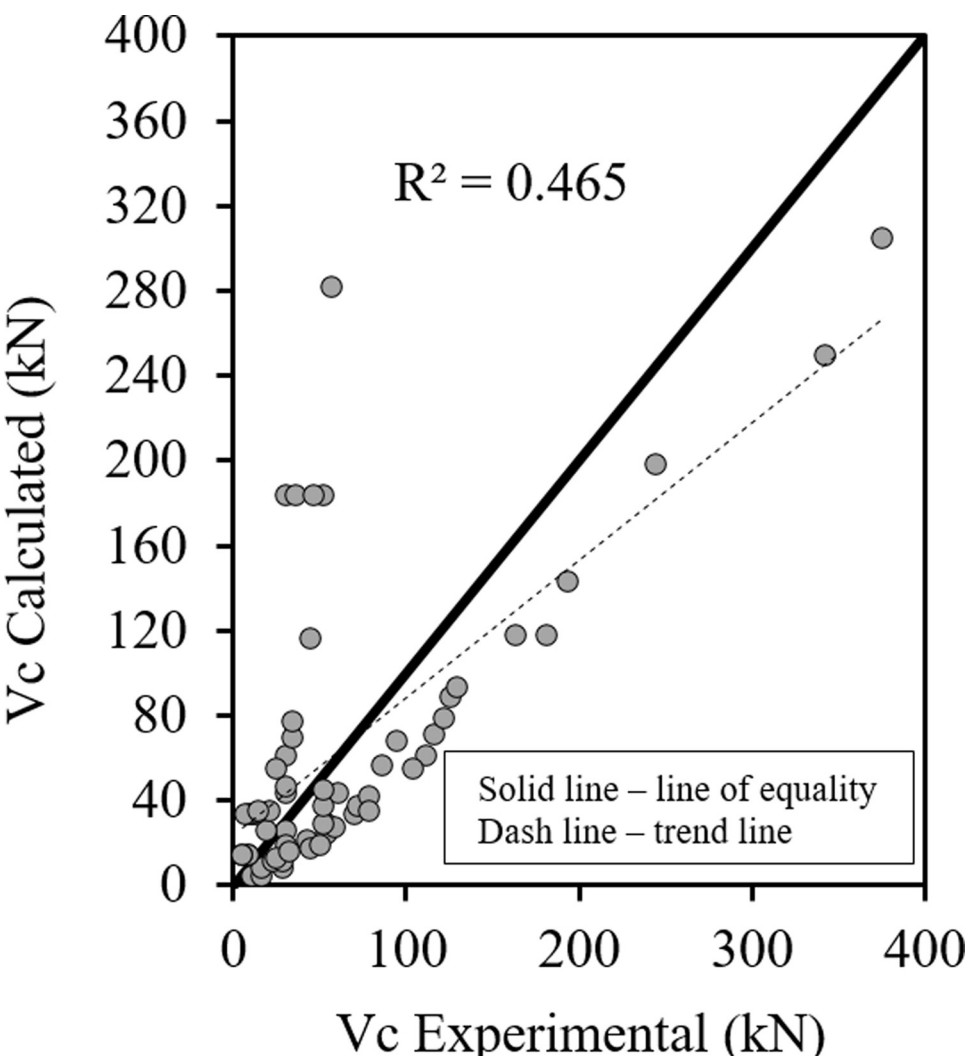

**Fig 8. Shear strength of fibrous HSC beams (Eq 11).**

Figs 8 and 9 show the relationship between the predicted results of Eqs 11 and 12 respectively versus the experimental data, as shown the results are close enough and in good correlation, also all data points are near the middle line and the statistical results are shown in Table 5.

A large database of experimental results is found in previous studies and used in this research (184) experimental tests found in the literature for HSC beams without steel fiber and (61) tests for HSC beams with steel fiber.

The concrete shear strength ($V_c$) of HSC beams is a function of the concrete compressive strength, longitudinal flexural reinforcement, and shear-span ratio. The result of the analysis shows that the shear strength of high strength concrete beams is proportional to the concrete compressive strength of power about 0.1, while the most available equation in codes and standards, the shear strength is proportional to ($f'_c$) of the power (0.5) as seen in Fig 10.

The multilinear and nonlinear regression analysis are used to propose different models for predicting shear strength ($V_c$) of HSC concrete beams, in terms of the main variables mentioned above. The predicted results from the proposed equations are compared with (26)

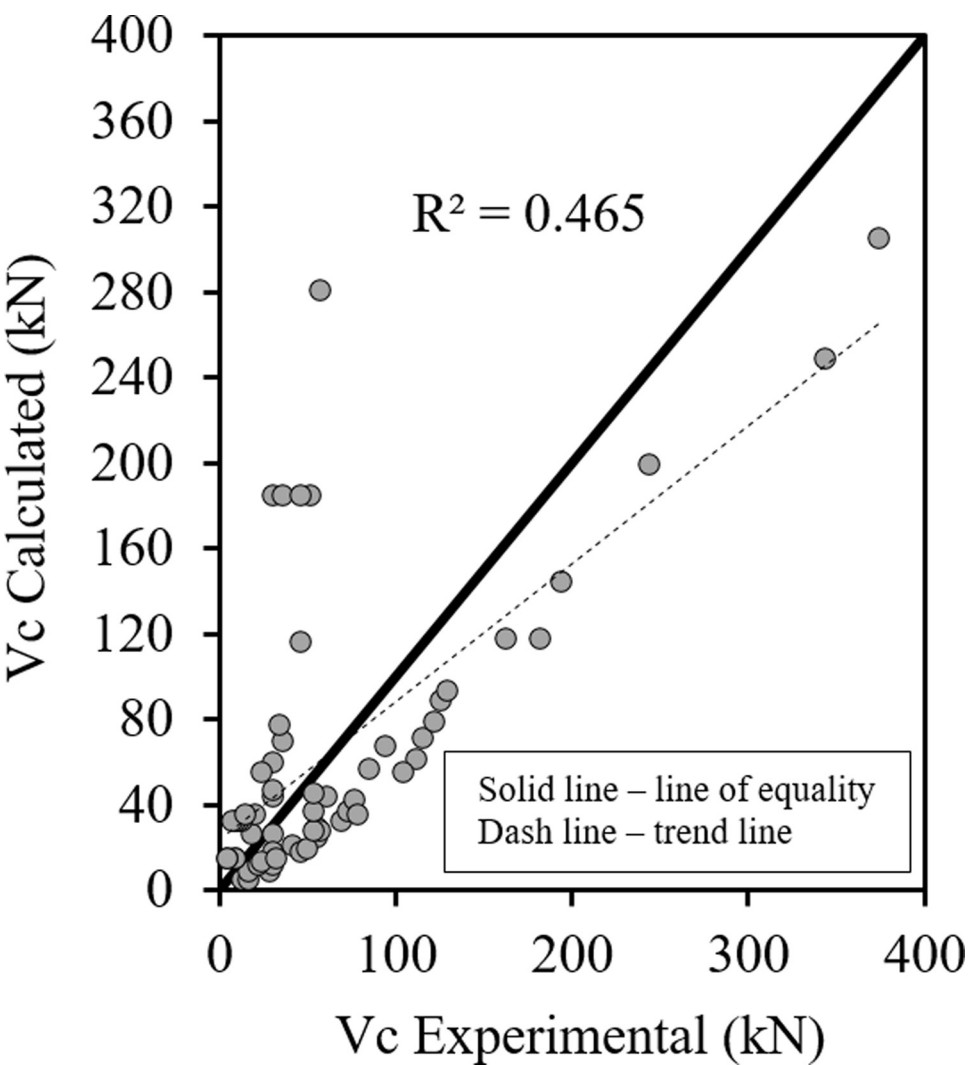

**Fig 9. Shear strengh of fibrous HSC beams (Eq 12).**

available equations found in previous studies, codes, and standards. As shown in Fig 10, equations [27–30, 34–36, 39–41, 43, 45, 47, 48] underestimated and equations [3, 37, 42, 49, 50] overestimated the shear strength. However, some equations [12, 28, 31, 33, 38] seem that they relatively have a good compatible unity but their $R_{ave}$ are still not close to one as well as their variation value is very high. While, the proposed models (EP), namely PE3 (Eq 8) and PE4 (Eq 9) are very compatible to unity with low variation value.

The predicted results from the proposed equations are better than the most available equations in literature, codes, and standards, and showed better correlations, and are very close to the experimental data (Fig 10).

**Table 5. Statistical results used in Eqs 11–12.**

| Equation | r | Var. | $\sigma$ | $R_{avg}$ | $R_{max}$ | $R_{min}$ |
|---|---|---|---|---|---|---|
| 11 | 0.774 | 0.324 | 0.57 | 1.1875 | 2.84 | 0.177 |
| 12 | 0.67 | 0.327 | 0.572 | 1.185 | 2.69 | 0.108 |

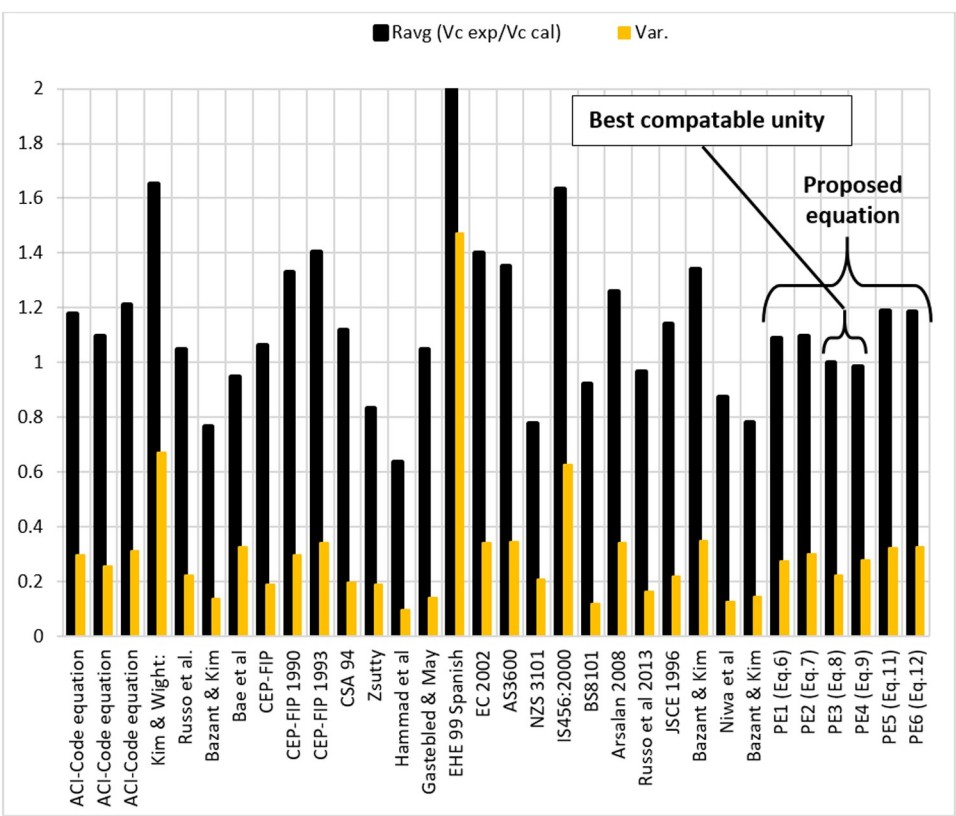

**Fig 10. Statistical results of the available equations for predicting shear strength of HSC beams.**

## 7. Conclusions

To predict shear strength ($V_c$) of HSC beam with and without steel fibers, a large database of experimental results is collected in previous studies and applied on 26 available equations found in previous studies, codes, and standards.

The shear strength of HSC beams is proportional to the concrete compressive strength of power about 0.1, while the most available equation in codes and standards, the shear strength is proportional to ($f_c'$) of the power (0.5).

The results show that most of the proposed models predict the shear strength of the HSC beam in away either it is overestimated or underestimated. Nevertheless, few models relatively have a good compatible unity but their $R_{ave}$ ($\frac{V_{excpermental}}{V_{calculated}}$) are still not close to one. Additionally, their variation value is very high. Thus, the multilinear and nonlinear regression analysis are used to propose different models for predicting $V_c$ of HSC concrete beam accurately. The proposed models are very compatible to unity with low variation value. In addition, the new models proposed by this study are including the fiber factor (F) which represents the fiber properties (fiber content $Q_f$%, fiber aspect ratio $\frac{L}{d}$, and fiber type) in the equation of shear strength in addition to other variables considered in this study. Apart from the proposed models, the shear behavior of HSC is complicated and not well-understood. That is why more research is necessary to study its performance in shear resistance and to find the best equation to predict it.

## Supporting information

**S1 Table. Experimental database for shear strength of HSC beams.**
(DOCX)

**S2 Table. Experimental database for shear strength of HSC beams with steel fiber.**
(DOCX)

## Author Contributions

**Conceptualization:** Ayad Zaki Saber.

**Data curation:** Ayad Zaki Saber.

**Formal analysis:** Ayad Zaki Saber.

**Funding acquisition:** Ayad Zaki Saber.

**Investigation:** Ayad Zaki Saber.

**Methodology:** Ayad Zaki Saber.

**Project administration:** Ayad Zaki Saber.

**Resources:** Ayad Zaki Saber.

**Software:** Ayad Zaki Saber.

**Supervision:** Ayad Zaki Saber.

**Validation:** Ayad Zaki Saber.

**Visualization:** Ayad Zaki Saber.

**Writing – original draft:** Ayad Zaki Saber.

**Writing – review & editing:** Ayad Zaki Saber.

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
