## [Decision Letter · Decision Letter 0]

19 Nov 2021

PONE-D-21-32899Prediction and developing of shear strength of reinforced high strength concrete beams with and without Steel Fibers Using Multiple Mathematical ModelsPLOS ONE

Dear Dr. Saber,

Thank you for submitting your manuscript to PLOS ONE. After careful consideration, we feel that it has merit but does not fully meet PLOS ONE’s publication criteria as it currently stands. Therefore, we invite you to submit a revised version of the manuscript that addresses the points raised during the review process.

We look forward to receiving your revised manuscript.

Kind regards,

Tianyu Xie, Ph.D.

Academic Editor

PLOS ONE

Journal Requirements:

"The funders had no role in study design, data collection and analysis, decision to

publish, or preparation of the manuscript"

"The authors have declared that no competing interests exist.

Powered"

Reviewers' comments:

Reviewer's Responses to Questions

**Comments to the Author**

1. Is the manuscript technically sound, and do the data support the conclusions?

Reviewer #1: Yes

Reviewer #2: Yes

2. Has the statistical analysis been performed appropriately and rigorously? 

Reviewer #1: Yes

Reviewer #2: Yes

3. Have the authors made all data underlying the findings in their manuscript fully available?

Reviewer #1: Yes

Reviewer #2: Yes

4. Is the manuscript presented in an intelligible fashion and written in standard English?

Reviewer #1: Yes

Reviewer #2: Yes

5. Review Comments to the Author

Reviewer #1: The authors have clearly answered my questions. In the reviewer's opinion, this manuscript can be accepted in the current form.

The following two papers are provided for the authors for reference:

[1] Yong Yu, Xinyu Zhao, Jinjun Xu, Cheng Chen, Simret Tesfaye Deresa, Jintuan Zhang. Machine learning‐based evaluation of shear capacity of recycled aggregate concrete beams. Materials, 2020, 13(4552).

[2] Xie T. Y., Aliakbar G., Togay O. Toward the development of sustainable concretes with recycled concrete aggregates: Comprehensive review of studies on mechanical properties. Journal of Materials in Civil Engineering, 2018, 30(9), 04018211.

Reviewer #2: 1. In Section 2, the author lists many research results, but only makes a brief statement and summary of each researcher's conclusions without any critical review or analysis of their progress or limitations. Reviewers believe that the Section 2 should have more discussion rather than a simple statement.

2. In Section 4, the author has collected a large amount of data, what criteria are used to select these data? please give more information and explanation to ensure the authenticity and reliability of these data.

3. Some figures do not highlight the advantages of the model obtained in this study over other models. It is suggested that the author make appropriate modifications to these figures (such as Fig.10). In addition, there are many statements irrelevant to the research of this paper in the conclusion part. Please make appropriate modifications to highlight the innovations of this research.

6. PLOS authors have the option to publish the peer review history of their article (what does this mean?). If published, this will include your full peer review and any attached files.

Reviewer #1: No

Reviewer #2: No

---

## [Author Response · Author response to Decision Letter 0]

11 Feb 2022

Dear reviewers and editor,

Enclosed please see the revised version of the manuscript as well as the response to your valuable remarks.

---

## [Decision Letter · Decision Letter 1]

7 Mar 2022

Prediction and developing of shear strength of reinforced high strength concrete beams with and without Steel Fibers Using Multiple Mathematical Models

PONE-D-21-32899R1

Dear Dr. Saber,

We’re pleased to inform you that your manuscript has been judged scientifically suitable for publication and will be formally accepted for publication once it meets all outstanding technical requirements.

Kind regards,

Tianyu Xie, Ph.D.

Academic Editor

PLOS ONE

Additional Editor Comments (optional):

Reviewers' comments:

Reviewer's Responses to Questions

**Comments to the Author**

1. If the authors have adequately addressed your comments raised in a previous round of review and you feel that this manuscript is now acceptable for publication, you may indicate that here to bypass the “Comments to the Author” section, enter your conflict of interest statement in the “Confidential to Editor” section, and submit your "Accept" recommendation.

Reviewer #1: All comments have been addressed

2. Is the manuscript technically sound, and do the data support the conclusions?

Reviewer #1: Yes

3. Has the statistical analysis been performed appropriately and rigorously? 

Reviewer #1: Yes

4. Have the authors made all data underlying the findings in their manuscript fully available?

Reviewer #1: Yes

5. Is the manuscript presented in an intelligible fashion and written in standard English?

Reviewer #1: Yes

6. Review Comments to the Author

Reviewer #1: I think that this paper can be accepted in its current form. The authors have well answered all of my questions.

7. PLOS authors have the option to publish the peer review history of their article (what does this mean?). If published, this will include your full peer review and any attached files.

Reviewer #1: No

---

## [Editor Report · Acceptance letter]

17 Mar 2022

PONE-D-21-32899R1 

Prediction and developing of shear strength of reinforced high strength concrete beams with and without Steel Fibers Using Multiple Mathematical Models 

Dear Dr. Saber:

I'm pleased to inform you that your manuscript has been deemed suitable for publication in PLOS ONE. Congratulations! Your manuscript is now with our production department. 

Kind regards, 

on behalf of

Dr. Tianyu Xie 

Academic Editor

PLOS ONE